# The New Materials for Battery Electrode Prototypes

**DOI:** 10.3390/ma16020555

**Published:** 2023-01-06

**Authors:** Rikson Siburian, Fajar Hutagalung, Oktavian Silitonga, Suriati Paiman, Lisnawaty Simatupang, Crystina Simanjuntak, Sri Pratiwi Aritonang, Yatimah Alias, Lin Jing, Ronn Goei, Alfred Iing Yoong Tok

**Affiliations:** 1Chemistry Department, Faculty of Mathematics and Natural Sciences, Universitas Sumatera Utara, Medan 20155, Indonesia; 2Carbon Research Center, Universitas Sumatera Utara, Medan 20155, Indonesia; 3Postgraduate Program, Department of Chemistry, Faculty of Mathematics and Natural Sciences, Universitas Sumatera Utara, Medan 20155, Indonesia; 4Physics Department, Faculty of Science, Universiti Putra Malaysia, Serdang 43400, Malaysia; 5Department of Chemistry, Faculty of Mathematics and Natural Sciences, Universitas Negeri Medan, Medan 20221, Indonesia; 6Faculty of Agriculture, Universitas Methodist Indonesia, Medan 20151, Indonesia; 7Department of Chemistry, Faculty of Science, Universiti Malaya, Kuala Lumpur 50603, Malaysia; 8University Malaya Centre for Ionic Liquids (UMCiL), Universiti Malaya, Kuala Lumpur 50603, Malaysia; 9School of Materials Science and Engineering, Nanyang Technological University, 50 Nanyang Avenue, Singapore 639798, Singapore

**Keywords:** graphite, graphene nano sheets, N-graphene nano sheets, electrolyte, Cu/GNS, battery electrode prototype

## Abstract

In this article, we present the performance of Copper (Cu)/Graphene Nano Sheets (GNS) and C—π (Graphite, GNS, and Nitrogen-doped Graphene Nano Sheets (N—GNS)) as a new battery electrode prototype. The objectives of this research are to develop a number of prototypes of the battery electrode, namely Cu/GNS//Electrolyte//C—π, and to evaluate their respective performances. The GNS, N—GNS, and primary battery electrode prototypes (Cu/GNS/Electrolyte/C—π) were synthesized by using a modified Hummers method; the N-doped sheet was obtained by doping nitrogen at room temperature and the impregnation or the composite techniques, respectively. Commercial primary battery electrodes were also used as a reference in this research. The Graphite, GNS, N—GNS, commercial primary batteries electrode, and battery electrode prototypes were analyzed using an XRD, SEM-EDX, and electrical multimeter, respectively. The research data show that the Cu particles are well deposited on the GNS and N—GNS (XRD and SEM—EDX data). The presence of the Cu metal and electrolytes (NH_4_Cl and MnO_2_) materials can increase the electrical conductivities (335.6 S cm^−1^) and power density versus the energy density (4640.47 W kg^−1^ and 2557.55 Wh kg^−1^) of the Cu/GNS//Electrolyte//N—GNS compared to the commercial battery (electrical conductivity (902.2 S cm^−1^) and power density versus the energy density (76 W kg^−1^ and 43.95 W kg^−1^). Based on all of the research data, it may be concluded that Cu/GNS//Electrolyte//N—GNS can be used as a new battery electrode prototype with better performances and electrical activities.

## 1. Introduction

Recently, graphene material has attracted the attention of researchers due to its high charge mobility (up to 200,000 cm^2^Vs). The most common method to synthesize graphene is to conduct a mechanical exfoliation from graphite. However, the mechanical exfoliation method only produces a small amount of graphene [1]. An alternative method of graphite exfoliation is to form graphene through a chemical process [2]. The chemical reduction of graphene oxide and flash nitrogen can produce large single-layered graphene with a relatively low cost [3,4].

The use of graphene as a building material for battery components can improve the electrical properties and provide an excellent chemical stability to the battery [5]. Graphene has high electrical and thermal conductivity values that allow for the transport and mobility of metal ions and electrons within its structure [6,7]. However, graphene has a unique electronic structure where the valence and conduction bands of graphene touch at the cones of the hexagonal Brillouine zone (K point) which forms a Dirac point so that graphene has no energy gap (band gap). It is thus difficult to be utilized directly as a semiconductor material [8,9]. Therefore, it is necessary to modify the graphene structure in order to improve its performance.

The defect of the graphene structure is an attempt to change the electronic structure and increase the overall activity of graphene, especially in terms of the electron transfer to and from graphene [10]. The introduction of defects in the graphene structure can be done by using dopants such as nitrogen, hydrogen, boron, and sulfur. Nitrogen is the most commonly used dopant material because it has an atomic size which is similar to C and also the atomic mass of N is closest to carbon and the electron-rich nature of the N atom [11,12,13]. By inserting nitrogen dopants into the graphene structure, it causes the interaction and bonding between the N atom and carbon atom (C—N); the N atom can donate its electrons to the graphene system. Consequently, the N atom can affect the local density state around the Fermi energy level of N-graphene [14]. Interestingly, the GNS and N—GNS can also be synthesized by large-scale and facile methods and they exhibit outstanding properties as support materials [15,16,17].

Pure GNS and N—GNS still have a lower electrical conductivity when compared to their alloys with metallic elements. In order to improve their conductivities, GNS and N—GNS can be combined with metals such as Mg, Ni, Ti, Cu, Zn, Sc, and Fe. Graphene has a large specific surface area (up to 2600 m^2^ g^−1^) so that it can act as a support material for metal particles [18,19]. Graphene sheets can minimize the occurrence of a metal particle agglomeration and are able to control the size distribution and distribution of metal particles deposited on their sheet’s surface [20,21]. In addition, the interactions that occur between the metal particles and graphene can improve and modify the electronic and magnetic properties of the graphene composites.

Furthermore, the addition or use of electrolyte materials into the electrochemical system of the battery may assist the mobility of ions and electron transport that take place in the battery’s external circuit system. Electrolytic materials can optimize the reducing power of metals on the surfaces of the GNS and N—GNS by controlling the mobility of electrons flowing through the structures of the GNS and N—GNS, so that the rate of energy consumption and battery usage will also be slightly longer [22].

Several methods of combining metals with graphene which are over reported are hydrothermal [23], microwave-assisted [24], self-assembly [25], and electrolysis [26]. Cu metal is selected as the particle embedded in C—π because of its considerable abundance in nature, low cost, and its excellent electrical and thermal conductivity. CuO/Graphene composites would be suitable for electrode material for lithium-ion batteries with an initial coulombic efficiency (68.7%) and reversible capacity of 583.5 mAh g^−1^, with 75.5% retention of the reversible capacity after 50 cycles [27]. In this research, synthesized GNS was mixed with the precursor of Cu by using impregnation method to produce a Copper—GNS composite (Cu/GNS). Cu/GNS composite have a higher electrical and thermal conductivity than copper [28].

As a result, a primary battery (fuse battery) with an integrated anode/electrolyte/cathode is required. This battery will be able to conserve resources and extend their life. This research developed a prototype-integrated primary battery using Cu/GNS//electrolyte//C—π (graphite, GNS, and N—GNS). Each prototype’s performance as a main battery was evaluated using an electrical multimeter, XRD, and SEM-EDX.

## 2. Materials and Methods

### 2.1. Materials Preparation

The materials used in this research are graphite commercial powder (Carbon 98 wt %, fly ash 2 wt %), sulfuric acid (H_2_SO_4_ 98 wt %), sodium nitrate (NaNO_3_ 99 wt %), potassium permanganate (KMnO_4_ 99 wt %), hydrogen peroxide (H_2_O_2_ 30 wt %), ammonia (NH_3_ 25 wt %), copper (II) chloride dihydrate (CuCl_2_ 2H_2_O 99 wt %), manganese dioxide (MnO_2_ 98 wt %), and ammonium chloride (NH_4_Cl 99 wt %). All the chemical reagents are used as received without any further purification.

### 2.2. Synthesis of Graphene Nano Sheets

Graphene Nano Sheets (GNS) were synthesized by using the modified Hummer method. A total of 0.2 g of graphite powder was put into a 1000 mL erlenmeyer flask, followed by the addition of 0.2 g of NaNO_3_ and 15 mL of H_2_SO_4_ 96%. The solution was then stirred for 2 h. Furthermore, the Erlenmeyer flask containing the mixture was placed in an ice water bath, 1 g of KMnO_4_ was added gradually, and the mixture was further stirred for 24 h. After stirring, 20 mL of H_2_SO_4_ 5% and 1 mL of H_2_O_2_ 30% were added to the solution, followed by stirring for 1 h. Then, the solution was centrifuged at 6500 RPM for 15 min to separate the filtrate and supernatant. Then, the filtrate was decanted from the solution. Furthermore, 25 mL of distilled water was added to the solution and stirred using a centrifuge at a speed of 6500 RPM for 15 min. The solution was transferred to a beaker glass, ultrasonicated for 5 h, then allowed to cool to produce a graphene oxide solution. A total of 100 mL of the resulting graphene oxide solution was added to 5 mL of ammonia (NH_3_ 25%), which was stirred for 72 h. Then, the mixture was filtered and dried at 100 °C for 2 h to obtain the GNS.

### 2.3. Synthesis of N—Graphene Nano Sheets (N—GNS)

N-Graphene Nano Sheets (N—GNS) were synthesized by using the nitrogen dopant method at room temperature. A total of 1 g of the resulting graphene powder was added with 10 mL of 10 M ammonia (NH_3_ 25%), or until the powder was fully submerged. The mixture was stirred for an additional 48 h at room temperature. Then, the mixture was filtered and dried at 100 °C to obtain the N—GNS powder.

### 2.4. Synthesis of Copper/Graphene Nano Sheets (Cu/GNS)

The synthesis of the anode material of the Copper/Graphene Nano Sheets (Cu/GNS) was carried out by the impregnation method. A total of 0.5 g of Graphene Nano Sheets powder and 0.1388 g of the Cu precursor (CuCl_2_ 2H_2_O crystals) were mixed and stirred for 1 h. Then, they were filtered using Whatman filter paper No. 42. The precipitate obtained was dried at 100 °C for 12 h, producing a black solid powder. The solid powder was weighed and stored for a further characterization.

### 2.5. Electrolyte

A total of 0.5 g of ammonium chloride (NH_4_Cl) was dissolved into 10 mL of distilled water to obtain the electrolyte solution. Then, 0.5 g of manganese dioxide (MnO_2_) was dissolved in 10 mL of distilled water.

### 2.6. Preparing Primary Battery Prototype

The preparation of Primary Battery Prototype Cu/GNS//Electrolyte//C—π was done by the combination of its constituents (Graphite, Graphene Nano Sheets, and N—Graphene Nano Sheets) in the ratio of 1:1:1. The primary battery prototype was prepared by using graphite as a cathode and Cu/GNS as an anode connected into two different containers. Then, both of them were added to the NH_4_Cl solution, where it served as an electrolyte, and added MnO_2_ solution, that served as a depolarization, which was then left to stand for 6 h. Then, both solutions were mixed and left to stand for 48 h so that both electrode mixtures reacted well, which was measured by the production of electrical energy. The Cu/GNS//Electrolyte//Graphite mixture was then filtered using Whatman No. 42 filter paper. The obtained Cu/GNS//Electrolyte//Graphite deposits were dried at 100 °C for 12 h to produce a solid powder, then they were labeled as Cu/GNS//Electrolyte//Graphite. Similar procedures were carried out to produce the Cu/GNS//Electrolyte//GNS and Cu/GNS//Electrolyte//N—GNS.

### 2.7. Materials Characterization

X-ray Diffraction (XRD) analysis was conducted using beam sizes of 10 mm × 10 mm, a Cu/Kα monochrome graphite radiation (λ = 1.5406) at 40 kV and 100 mA. The range of 2θ is from 10° to 90° in a 2.0° step. The detection of the powder X-ray Diffraction patterns was done using the SWXD Diffractometer, Rigaku Corporation (Singapore), which operates at 18 kW and collects two-dimensional diffraction patterns from the sample. The data were processed using D/MAX—2000/PC version 3.0.0.0, while the SEM-EDX analysis was conducted using EM 30 COXEM with an accelerating voltage of 20 kV.

### 2.8. Measurement Electrical Conductivity

The measurement of the electrical conductivity of the prototype material Cu/GNS//Electrolyte//C—π (Graphite, GNS, and N—GNS) was performed using an electrical multimeter. A total of 0.25 g of Cu/GNS//Electrolyte//Graphite powder was inserted into the glass fuse and was compacted until it was completely covered by the fuse cover. The device was then connected to the Multimeter and DC Power Supply circuit. The electrical conductivity of the Cu/GNS//Electrolyte//Graphite was measured at an applied voltage of 0.5, 1.0, and 1.5 Volts. The electrical conductivity of the Cu/GNS//Electrolyte//GNS and Cu/GNS//Electrolyte//N—GNS powders was measured using the same procedure.

## 3. Results

### 3.1. Synthesis Graphene Nano Sheets

Figure 1 shows the XRD diffraction pattern of the Cathode Prototype Graphite, GNS, and N—GNS (C—π), which shows that there is a peak difference at 2θ = 26.5° for C (002). For the graphite, there is a sharp and narrow peak at 2θ = 26.5°, indicating a d-spacing value of 3.33 Å [29].

### 3.2. Characterization Material Electrode Primary Battery Prototype without Electrolyte and Electrode Commercial Primary Battery Prototype

A prototype electrode diffractogram of a primary battery without electrolytes and a commercial battery is shown in Figure 1.

Figure 1 shows the XRD diffraction pattern of the commercial primary battery anode (zinc plate); there is a sharp, tight peak at 2θ = 43°, indicating that the Zn metal (101) is the main component of the commercial primary battery anode. In the cathode of the commercial primary batteries, the diffractogram is similar to graphite. In the XRD diffraction pattern of the Cu/GNS prototype anode, there is a weak and narrow peak at 2θ = 42.66°, indicating that the Cu atom (111) is well deposited in the GNS [30].

Then, the Scanning Electron Microscope (SEM) morphology images of Graphite, GNS, N—GNS, and the electrodes of the commercial battery are presented in Figure 2.

Figure 2a shows the SEM image of the commercial primary battery anodes, which consist of a blob-shaped surface, clumps which have accumulated and which are the shape of particles, and particles which can be likened to piles of needles. There are many circle white spots embedded in the graphene surfaces of the anode (Figure 2a) and (Figure 2c). The Cu/GNS SEM image shows that there are white spots shaped like ovals on the surface of the GNS, which proves that Cu was successfully deposited on GNS (Figure 2c). This means that there are many metals which have been well deposited on the GNS. These data are consistent with the XRD data. On the surface of the commercial primary batteries’ cathode, flake-shaped solids can be seen to be piled up (Figure 2b). The SEM image of the commercial battery cathodes show that the structure is arranged by a stack of structures and is very tight. The surface of the graphite is stacked in the form of flakes (Figure 2d). Interestingly, the surface GNS is very different from the Graphite (Figure 2e).

### 3.3. Electrical Conductivity

The electrical conductivity comparison of prototype and commercial electrodes are presented in Figure 3.

The electrical conductivity numbers of graphite measured at 5–30 V have a stable electrical conductivity (Figure 3d). Interestingly, the electrical conductivity of the GNS and N—GNS are higher compared to graphite (Figure 3e,f). In addition, the GNS may affect the electrical conductivity of Cu on Cu/GNS (Figure 3c). Unfortunately, the GNS, N—GNS, and Cu/GNS have the lower electrical conductivity values among both commercial electrodes.

### 3.4. Power Density vs. Energy Density

The power density vs. energy density data show the ratio of stored energy and the transfer of the energy rate in the material (Figure 4).

The power density vs. energy density is measured in order to evaluate the performance materials which act as the cathode and anode. Figure 4a,b show that the power density vs. the energy density values of both commercial battery electrodes are stable. Their patterns are quite similar patterns with graphite; the GNS and N—GNS and totally different compared to the Cu/GNS (Figure 4c–f).

### 3.5. Characterization Material Primary Battery Prototype with Electrolyte

The SEM images of the Cu/GNS//Electrolyte//Graphite, Cu/GNS//Electrolyte//GNS, and Cu/GNS//Electrolyte//N—GNS are presented in Figure 5a–d.

We found the many of the white circle-shaped spots distributed on the surfaces of the GNS and N—GNS (labelled in a red circle) were probably Cu metals on C—π. Interestingly, the SEM images obviously show the existence of both electrolyte and cathode, which clearly appear on battery prototype (Cu/GNS//Electrolyte//C—π). This means that there is a good interaction among the anode material//electrolyte//cathode material on the battery prototype. Subsequently, we carried out the SEM mapping and EDX in order to make sure that Cu exists on the C—π surfaces (Figure 6).

Figure 6 shows that there are white spots with small circles on the Cu/GNS//Electrolyte//Graphite, indicating that the Cu particles were successfully deposited on the GNS surface. In the SEM micrograph of the prototype, visible Cu particles are scattered evenly on the surface of the GNS.

Further, the weight concentration element prototype can be seen in Table 1.

Table 1 also clearly shows that Cu is well deposited in the GNS (the Cu element is 20.24% *w/w*). The primary battery prototype of the Cu/GNS//Electrolyte//GNS has an EDX of the Cu elements (2.21% *w/w*) which was greater than Cu at the Cu/GNS//Electrolyte//N-GNS (1.94% *w/w*). The relationship between the particle size and variations in the Cu concentration for different prototypes can be seen in Figure 7.

The particle size of Cu was estimated by SEM. The particle size distribution of Cu is shown in Figure 7 for the Cu/GNS prepared with the different material types of the cathode on the primary battery prototype. The mean sizes of Cu were estimated by analyzing a few randomly chosen areas containing 250 particles (white spot in red circle) (Figure 5a–d) in magnified SEM images. Then, the SEM images were calculated and analyzed with the histogram. Finally, we summarized them in Figure 7. It is found that the Cu particles were well dispersed on the GNS. The average diameters of Cu were 0.43–0.58 µm for the Cu/GNS catalyst prepared at different cathode materials (Graphite, GNS, and N—GNS). This method was referred to in our previous study [20].

Figure 7 shows that the Cu particle size decreases at the Cu/GNS to Cu/GNS//Electrolyte//Graphite prototype, then significantly increased at the Cu/GNS//Electrolyte//GNS, and increased slowly until the Cu/GNS//Electrolyte//N–GNS. Those data show the support material (C—π) may affect the performance of Cu, especially the particles sizes of Cu.

### 3.6. Electrical Conductivity of Prototypes with Electrolyte

The electrical conductivity of the Cu/GNS//Electrolyte//GNS prototype is higher than with the Cu/GNS//Electrolyte//N—GNS, Cu/GNS//Electrolyte//Graphite, and commercial batteries (Figure 8).

This is because where graphene has an excellent electron mobility by being added with electrolytes, its electrical conductivity will be very stable. The addition of Cu particles deposited in the GNS further enhances the electrical conductivity of the Cu/GNS//Electrolyte//GNS prototype.

### 3.7. Power Density vs. Energy Density of Prototypes with Electrolyte

The power density vs. energy density shows the comparison of the energy stored and the transfer rate of the energy in the battery. The summary of the power and energy density of the prototype batteries in the presence of the electrolyte is shown in Figure 9.

## 4. Discussion

Firstly, we characterized our GNS and N-GNS products with XRD (Figure 1). A weak and widening peak at peak 26.5° indicates the formation of Graphene Nano Sheets (GNS). The presence of the peak of the GNS is due to the bonding of the oxidized graphite and the introduction of oxygen into the interlayer chamber in the graphite [29] as well as the reduction in the oxygen functional groups because of the presence of ammonia to produce GNS. A slightly sharp and widened peak at the cathode of 2θ = 26.5° indicates that GNS interacting with the nitrogen from ammonia produces N-GNS. This is characterized by the exfoliation of graphene with the introduction of nitrogen atoms from ammonia into the graphene structure [31,32]. Further, the GNS and N-GNS were analyzed with an SEM measurement. The structure of the graphene looks like a sheet and does not appear to be a hexagonal structure, which means that the graphene is successfully synthesized. In contrast, the surface of N—GNS shows that the surface of N—GNS (Figure 2f) looks very different from the commercial battery, graphite, and graphene, where the surface looks more encased than graphene. That is, the N atom interacts with the structure of graphene. In order to clarify this further, the electrical conductivity data show that the electrical conductivity of the commercial electrodes is still higher than the prototype electrodes. This is because commercial electrodes have been added with electrolytes, resulting in a higher electrical conductivity when compared with the prototype electrodes that have not been added with electrolytes. The added electrolytes will improve the mobility of the electrons so that the electrical current delivery power is stable. Clearly, the stacked graphite structure may affect the lower electrical conductivity number compared to the GNS and N—GNS. The thin and flat layers of the GNS and the existence of the N atom on the N—GNS may improve the electrical conductivity number. It means that the structure of the material may affect the properties of the material. This is consistent with the electrical conductivity data of the Cu/GNS, which suggests that the present GNS may change the electrical conductivity of Cu (Figure 3). All of the electrical conductivity data are consistent with the power density data (Figure 4). The anode and cathode of the primary battery are dominant, containing Zn and C (graphite), respectively. The Zn metal and C (graphite) show the stable pattern concerning the power density vs. the energy density data. It means that Zn is expected as the electron source in the primary battery and that C (graphite) stores electrons well, thus an oxidation–reduction reaction should occur. The power density vs. the energy density value of the GNS is higher than a commercial electrode. This is due to the higher mobility of the electrons and larger surface area of the GNS so that the transfer of the electrons is faster and more electrons are stored, resulting in a higher power density and energy density.

In Figure 5 and Figure 6, it may be seen that the sheets of graphene are arranged regularly within a thin layer. In the SEM micrographs of the primary battery prototype Cu/GNS//Electrolyte//N—GNS, it can be seen that the Cu metal’s particles are deposited into the N—GNS. The spread of the Cu particles is evenly distributed, but there is a stack between one particle and another particle so that it forms a cluster on the surface of N C—π GNS. Then, the date of the particle sizes may be seen in Table 1 and Figure 7. The data show that the Cu/GNS//Electrolyte//N—GNS have the highest particle size of 0.4338 µm and the Cu/GNS//Electrolyte//Graphite have the smallest particle size of 0.5794 µm. This is due to the number of copper atoms in GBN, each of which is different so that the particle size also varies.

In order to determine the effect of the electrolyte, we provided and evaluated the electrical performances of the prototypes of the Cu/GNS//Electrolyte//C—π (Figure 8 and Figure 9). Therefore, a deep evaluation of each of the electrode’s materials containing the battery is definitely required. Generally, the primary battery consists of the main parts, containing the anode (Zn), electrolyte (NH_4_Cl—MnO_2_), and cathode (C—graphite). In this research, we provide:

(i) An electrode materials candidate for a primary battery with and without an electrolyte, those are Cu/GNS, Graphite, GNS, and N—GNS.

(ii) A prototype of a primary battery: Cu/GNS//NH_4_Cl—MnO_2_//C—π (Graphite, GNS and N—GNS). All of these are then compared with the commercial primary battery.

The graphite and our product (GNS and N—GNS) may be expected to store electrons and interestingly modify the properties of the Cu metal. Note, the Cu metal is more difficult in releasing its electron compared to the Zn metal (voltaic series), meaning the role of the GNS is pivotal in order to modify the properties of Cu. The electrolyte has a very important role in the battery; the electrical properties of the battery are lower without the electrolyte. The electrolyte of NH_4_Cl—MnO_2_ may be used to improve the electricity of Cu/GNS and others of the C—π.

They exhibit a higher power density vs. the energy density value than the Cu/GNS//Electrolyte//N-GNS, Cu/GNS//Electrolyte//Graphite, and commercial battery. This is due to two factors, namely (i) entropy. Graphene has a large surface area, so the Cu particles can be dispersed uniformly within the structure of Graphene. The second factor is (ii) Enthalpy (the chemical interactions between Cu and Graphene).

The pivotal points in order to improve the performance of the primary battery are (i) the non-stoichiometry releasing and storing of electrons between the anode and cathode, meaning the lifetime of the usage of the primary battery will be short, and (ii) the thermodynamically of the electrode materials of the primary battery, meaning the properties of the material linearly affect the performance of the battery, such as the entropy and voltage values.

## 5. Conclusions

In this paper, we provide the material for a battery electrode base on C—π (Graphite, GNS, and N-GNS), electrolyte (NH_4_Cl—MnO_2_), and Cu/GNS and evaluate their performance with and without an electrolyte. We found that the Cu/GNS//Electrolyte//GNS has the highest electrical conductance number over the others. This means that it may be a candidate material for a battery electrode.

## Figures and Tables

**Figure 1 materials-16-00555-f001:**
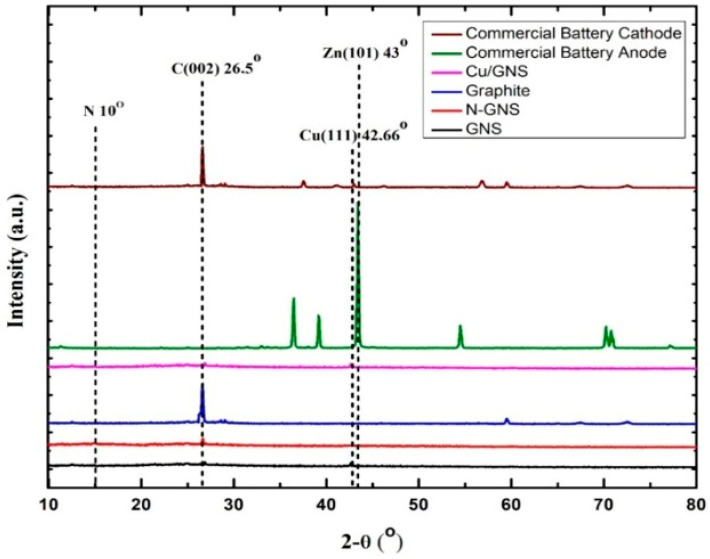
Diffraction pattern XRD anode and cathode primary battery prototype and commercial.

**Figure 2 materials-16-00555-f002:**
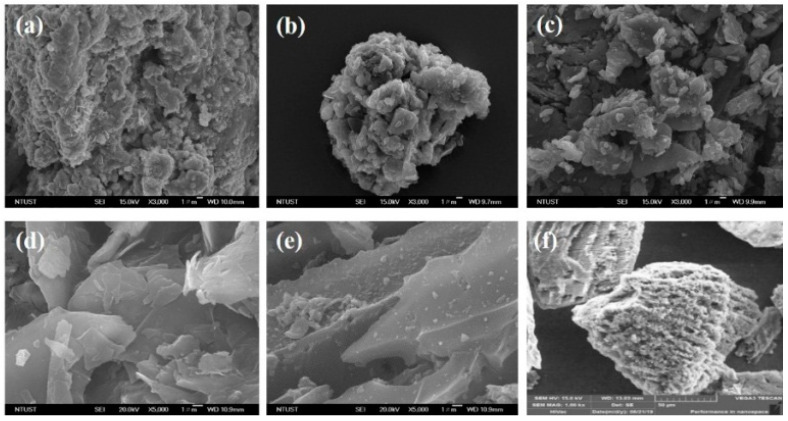
SEM images of (**a**) commercial battery anode, (**b**) commercial battery cathode, (**c**) Cu/GNS, (**d**) Graphite, (**e**) GNS, and (**f**) N—GNS.

**Figure 3 materials-16-00555-f003:**
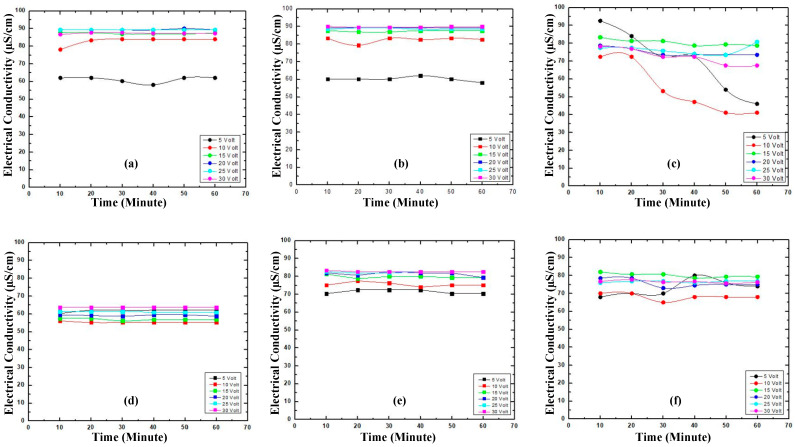
Electrical conductivity electrode (**a**) commercial battery anode, (**b**) commercial battery cathode, (**c**) Cu/GNS, (**d**) Graphite, (**e**) GNS, (**f**) N—GNS.

**Figure 4 materials-16-00555-f004:**
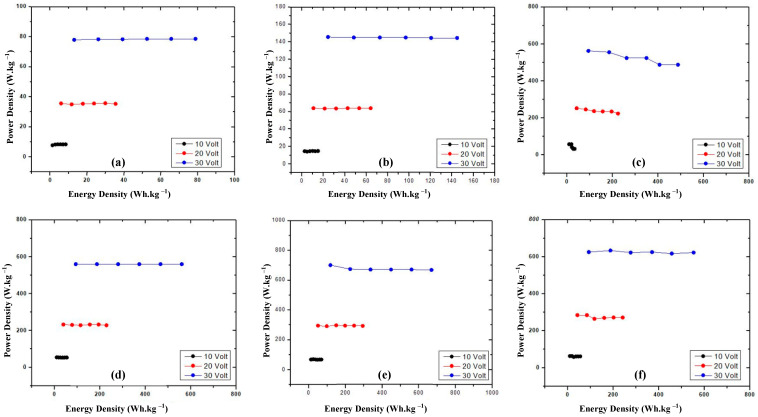
Power density vs. energy density electrode (**a**) commercial battery anode, (**b**) commercial battery cathode, (**c**) Cu/GNS, (**d**) graphite, (**e**) GNS, (**f**) N—GNS.

**Figure 5 materials-16-00555-f005:**
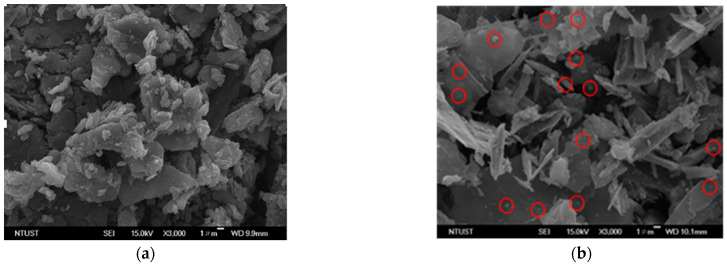
(**a**) SEM image of Cu/GNS, (**b**) SEM image of Cu/GNS//Electrolyte//Graphite, (**c**) SEM image of Cu/GNS//Electrolyte//GNS, (**d**) SEM image of Cu/GNS//Electrolyte//N—GNS. Red circles correspond to Cu particles.

**Figure 6 materials-16-00555-f006:**
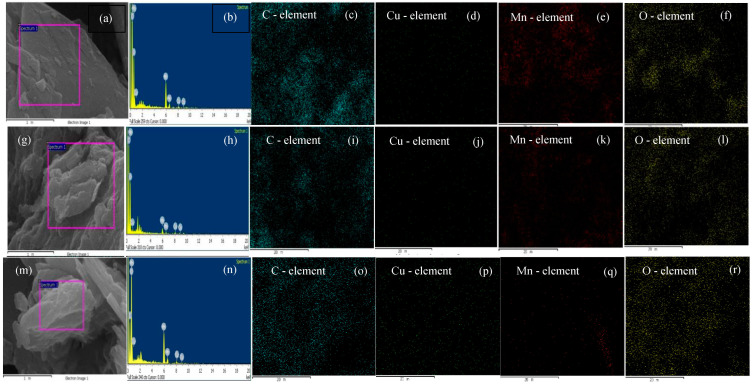
SEM image, EDX graph, C element mapping, Cu element mapping, Mn element mapping, and O element mapping from Cu/GNS//Electrolyte//Graphite (**a**–**f**), Cu/GNS//Electrolyte//GNS (**g**–**l**), and Cu/GNS//Electrolyte//N—Graphene (**m**–**r**).

**Figure 7 materials-16-00555-f007:**
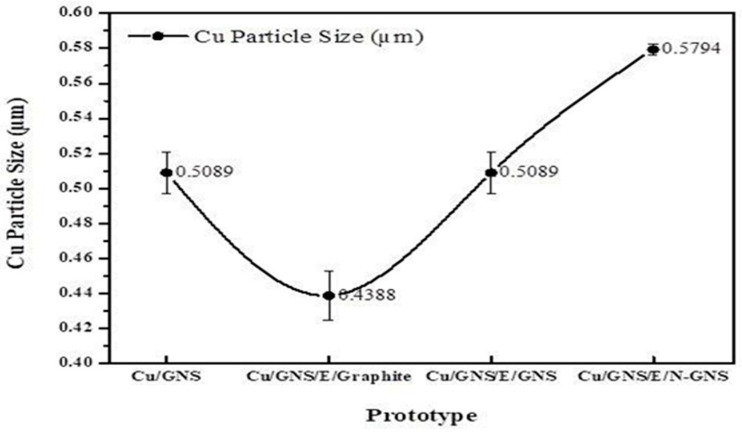
Cu particle size with primary battery prototype.

**Figure 8 materials-16-00555-f008:**
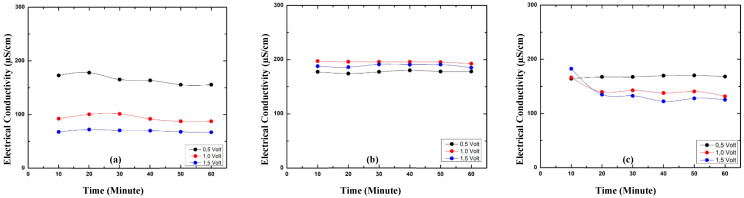
Electrical conductivity primary battery prototype (**a**) Cu/GNS//Electrolyte//Graphite, (**b**) Cu/GNS//Electrolyte//GNS, (**c**) Cu/GNS//Electrolyte//N—GNS.

**Figure 9 materials-16-00555-f009:**
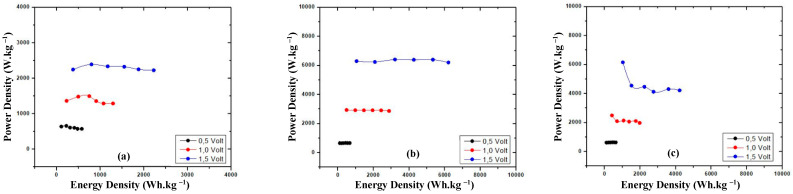
Power density vs. energy density primary battery prototype (**a**) Cu/GNS//Electrolyte//Graphite, (**b**) Cu/GNS//Electrolyte//GNS, (**c**) Cu/GNS//Electrolyte//N—GNS.

**Table 1 materials-16-00555-t001:** Weight concentration element prototype with used EDX data.

Sample	Weight Concentration Element (%)			
	C	O	Mn	Cu
Cu/GNS	44.31	34.75	0.70	20.24
Cu/GNS//Electrolyte//Graphite	47.27	26.63	25.65	0.45
Cu/GNS//Electrolyte//GNS	51.06	40.58	6.15	2.21
Cu/GNS//Electrolyte//N-GNS	24.65	39.56	33.84	1.94

## Data Availability

The data presented in this study are available from the corresponding authors upon reasonable request.

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
