# Peer review of "The New Materials for Battery Electrode Prototypes"

_materials, 2023, doi:10.3390/ma16020555_

Round 1

Reviewer 1 Report

The manuscript titled “The New Materials for Battery Electrode Prototypes” by Siburian et al investigates the performance of Copper (Cu)/Graphene Nano Sheets (GNS) and C-p(Graphite, GNS, and N-GNS) as a new battery electrode prototype. The results are interesting and the manuscript should be accepted with minor revision. Some comments that need to be taken care of before accepting the manuscript are:

1)      Introduction, paragraph 1 the laser and flash induced graphene is also a low-cost process and needs to be discussed.
https://doi.org/10.1021/acsami.2c09096

https://doi.org/10.1021/acsmaterialslett.2c00616

2)      Section 4. Discussion line 7 “In order to more proving” isn’t grammatically correct

3)      Fig.6 , the error bar for Cu size distribution needs to be shown to confirm the trend.

Author Response

University of Sumatera Utara

Department of Chemistry, Faculty of Mathematics and Natural Sciences

Jl. Bioteknologi, Padang Bulan, Medan, Sumatera Utara

Indonesia

21 December 2022

Dear Editor-In-Chief:

Materials

We would like to submit our revised manuscript Manuscript Title: “The New Materials for Battery Electrode Prototypes” Authors: Rikson Siburian et al to Materials.

We are very delightful on the Reviewer 1 comments. Thus, we revised our manuscript base on the Reviewers 1 and 2 comments and suggestions. Hopefully, it may be published now 

                                                                                                                        Sincerely yours,

Rikson Siburian, PhD

Department of Chemistry, Faculty of Mathematic and Natural Sciences

Jl. Bioteknologi, Padang Bulan, Medan, Sumatera Utara

Indonesia

Mobile: +62-812-3770-541

                   e-mail: rikson@usu.ac.id

Reviewer 2 Report

In this manuscript, Rikson Siburian et al. report the material for battery electrode base on C–π (Graphite, GNS and N-GNS), electrolyte (NH4Cl–MnO2) and Cu/GNS and evaluate their performance with and without electrolyte. The conclusion is supported by the present data. This work can offer some inspiration in this research field. I would recommend the major revision of this manuscript based on the following notes.

1. Abbreviations in the abstract should be written in a clear manner. For example, what is N-GNS?

2. SEM images show the morphologies of electrode materials, not the electrode. They are different. Please check in the whole manuscript.

3. The authors said that the SEM images of Cu/GNS//Electrolyte//Graphite, Cu/GNS//Electrolyte//GNS, and Cu/GNS//Elec-trolyte//N – GNS were presented in Figure 5. However, it is very confusing. What part do SEM images belong to? You can get some experience from these recent reports (Energy Stor. Mater., 2022, 46, 583; Small, 2022, 18, 2105185; Chem. Eng. J., 2022, 428, 131031). Besides, please provide the EDX mapping results to make the statistical results clear.

4. In the SEM micrograph of the prototype (Figure 5), visible Cu particles are scattered evenly on the surface of the GNS. However, I cannot see any Cu particles.

5. What method did you use to distinguish particle size (Figure 6)? Please make it clear.

6. There are many typing errors in the manuscript. Please check the manuscript carefully.

Author Response

University of Sumatera Utara

Department of Chemistry, Faculty of Mathematics and Natural Sciences

Jl. Bioteknologi, Padang Bulan, Medan, Sumatera Utara

Indonesia

21 December 2022

Dear Editor-In-Chief:

Materials

We would like to submit our revised manuscript Manuscript Title: “The New Materials for Battery Electrode Prototypes” Authors: Rikson Siburian et al to Materials.

We are very delightful on the Reviewer 1 and 2 comments. Thus, we revised our manuscript base on the Reviewer 1 comments and suggestions. Hopefully, it may be published now

                                                                                                                        Sincerely yours,

Rikson Siburian, PhD

Department of Chemistry, Faculty of Mathematic and Natural Sciences

Jl. Bioteknologi, Padang Bulan, Medan, Sumatera Utara

Indonesia

Mobile: +62-812-3770-541

                   e-mail: rikson@usu.ac.id
